# An Analysis of Food Waste in Czech Households—A Contribution to the International Reporting Effort

**DOI:** 10.3390/foods10040875

**Published:** 2021-04-16

**Authors:** Petra Nováková, Tomáš Hák, Svatava Janoušková

**Affiliations:** 1Faculty of Humanities, Charles University, 110 00 Prague, Czech Republic; novakova.petra.cz@gmail.com; 2Environment Centre, Charles University, 110 00 Prague, Czech Republic; 3Faculty of Science, Charles University, 110 00 Prague, Czech Republic; svatava.janouskova@czp.cuni.cz

**Keywords:** food waste, household waste, waste monitoring, kitchen diaries, sustainable development, Czech Republic

## Abstract

Food waste originating in households in the Czech Republic is an important but unknown issue. Due to the country’s membership among the most developed economies (European Union, OECD) and its commitments towards UN Sustainable Development Goals, the government must, inter alia, significantly reduce household food waste. However, reliable data and indicators based on internationally agreed approaches and methods have been missing so far. This article brings original results from a survey comprising over 400 Czech households based on the kitchen diaries method showing that, on average, surveyed households discarded 2.6 kg (1.1 kg per capita) weekly. After extrapolation, the total food waste was estimated to be 135.7 kg per household or 57.1 kg per capita annually. Half of the total food waste from surveyed households was thrown into municipal bins for mixed waste. Despite a relatively low total food waste stream, its disposal needs substantial improvement to meet national and international regulations as well as sustainability criteria.

## 1. Introduction

People have known the unique role of food from the start of civilization, but surprisingly world hunger is on the rise in the 21st century. An estimated 820 million people did not have enough to eat in 2018 (almost 11% of the global population suffer from chronic undernourishment) [1] and the world is not on track to achieve the second sustainable development goal, known as Zero Hunger, by 2030. At the current pace, approximately 37 countries will fail even to reach low hunger (inadequate food supply) as defined by the severity scale of the Global Hunger Index by 2030 [2]. At the same time, one-third of the total food the world produces, around 1.3 billion tons, is wasted yearly [3]. Besides suffering from hunger, undernourishment and subsequent health problem, there are significant environmental impacts associated with pesticide and fertilizer production and use, livestock wastes, soil and habitat degradation in the agricultural sector. Regarding climate change, the global food system, including food waste and loss, was estimated at 21–37% of the world total annual CO2_eq_ emissions for the period 2007–2016 [4], and an estimated 8–10 per cent of global greenhouse gas emissions are associated with food that is not consumed [5,6]. Just these two aspects, the impacts on human and environmental well-being, create an unequivocal moral imperative for reducing food waste at all levels. 

Although the Czech Republic (hereinafter Czechia, contributes to these problems, as do many other industrialized countries, politicians, managers, businesspeople and citizens do not know much about it [7]. The problem is not a lack of public support for necessary changes (over 40% of Czechs have regularly considered food wastage to be a serious problem [8]) but there are little data and information about the issue. There have been some unsystematic surveys and estimates on food wastage in schools or services organized by NGOs or interested businesses. For example, the Association of Social Responsibility supported the initiative “I love food, I do not waste” aimed at food wastage in school canteens. The campaign was based on estimates by school employees that 20–35% of cooked meals are finally wasted. In other words, about 48,000 tons of processed food yearly or 240 tons daily are disposed of by pupils and students at school canteens in a small country with a population of 10.6 million [9]. Czech restaurants and fast-food outlets also contribute significantly to the total food waste by producing about 27,000 tons of waste yearly, most of it at the processing phase [10]. 

Food management in Czech households thus seems to be an important but unknown issue. This article does not seek to formulate any hypothesis on the measured amount of food wastage by Czech households, since the economy has been completely transformed since 1989 (after the fall of communism). The economic performance of the country has increased by magnitudes of orders up until today: foreign trade turnover is twenty times higher, total GDP in CZK has increased by almost 700% in nominal terms and 75% in real terms. Besides economic progress, the overall quality of life has increased as well: the average life expectancy for both sexes has increased from 71.5 to 79.5 years, people consume much more fresh vegetables (66 to 100 kg per person a year) and fruit (60 to 86 kg per person a year), and healthy meat such as poultry (14 to 29 kg per person a year). Phenomena related to changed consumption and production patterns have also developed significantly: the yearly production of solid household waste has increased from about 220 kg (estimated) to 353 kg per person and the share of a household’s expenditure on food and eating has dropped from 34 to 22% [11,12,13]. 

The Czech Republic has undergone immense changes in all spheres of life during thirty years of its overall transformation. Since there have been no other data on food wastage from other post-communistic countries until recently (just Hungary [14]), several distinct results about household food waste might be expected: 

1. Low food waste. Since the older generation (over 65 years) spent half of their lives until 1989 living in a totalitarian regime with limited supplies including food, they keep their thrifty overall lifestyle (including food) resulting in a low amount of waste. Therefore, this group of people, accounting for 20% of the population [15], strongly influences the nation-wide food waste average data. This assumption might be also loosely supported by Sosna et. al., who investigated household food wastage in the rural environment in Czechia by combining waste composition analysis and ethnographic research in one village. They explained the low wastage of edible food (7.9 kg per capita a year) by a concept of thrift that includes economizing via self-denial or creative management of resources, moral discourse and social relations [16]. 

2. Moderate food waste. This possibility proceeds from the country’s socio-economic development since 1989: the country has experienced great progress in all spheres of life, belongs among the average European Union (EU) member states and, as an OECD member state, among highly developed countries globally (human development index, gross domestic product per capita, material footprint per capita) [17]. Thus, a reasonable assumption is that the Czech per capita food waste stream will be commensurate with those in peer countries (e.g., Malta, Spain, Portugal, Slovenia, Cyprus). 

3. High food waste. This is based on the objective of rapid growth and “catching up and overtaking” capitalism that played a central role in the formation and evolution of socialist systems [18]. In other words, the countries in postcommunist transformation must have inevitably gone through all the mistakes of more developed countries, such as consumerism, overconsumption and wasting resources, before reaching a mindset and systemic change of advanced (postmodern) capitalism featuring a philosophy of efficiency, sharing and circularity, amongst others. Thus, many Czechs still long for several large cars in the family garage, regardless of an efficient public transport system and overfilling their grocery shopping baskets regardless of real needs. 

To set a baseline situation with regard to food waste in an important economic sector, we analyze the food waste patterns of Czech households based on recent data collection. In the absence of Czech legislation or even nonbinding instruments, we use UN and EU commitments as a broad but important policy context for framing the first empirical results. The main contribution of the article is, however, the methodological and result sections bringing original findings on food wastage in Czech households. The applied approach proceeds from a brief review of the methods used in similar research. 

## 2. The Study Theoretical Background

Internationally, food losses and waste have received high recognition. The UN Agenda 2030, introduced and endorsed in 2015 as a 15-year plan of action for people and the planet, set 17 Sustainable Development Goals (SDGs) including measurement and boosting of progress in all dimensions of sustainable development. Target 12.3 explicitly calls for halving per capita global food waste at the retail and consumer levels and reducing food losses along production and supply chains by 2030. 

In 2011, the European Commission’s “Roadmap to a Resource Efficient Europe” provided a framework for future actions and milestones for resource efficiency to be met by 2020. The milestone related to food proposed that by 2020 "disposal of edible food waste should have been halved in the EU" [19]. Despite a lack of harmonized data and methods for analyses, it is assumed that about 20% of the food produced is wasted in Europe [20]. Therefore, the European Commission, besides monitoring SDGs goals and targets for the EU as a whole [21] and at the member state level, also continues to consider food waste as a key to achieving sustainability in its policy “Farm to Fork Strategy”, published in May 2020 [22]. It is at the heart of the European Green Deal aimed at making food systems fair, healthy and environmentally friendly [23]. This double challenge places high demands on member states, including the Czech Republic, on implementing effective measures with an instrumental role of data and information. 

### 2.1. Food Waste Monitoring 

Food may be lost during any phase of the food supply chain [24]. In developed countries, food waste mostly occurs in the final stages because of decisions made by retail, food services and household members, among others [25]. It is households that are responsible for the major share of the food wasted in developed countries [26,27]. Food waste in the consumption stage represents a real waste and incurred cost and financial loss for households. Besides this, the environmental impact of food production and processing occurs regardless of whether consumers, households in this case, use or waste food [28,29]. 

Global monitoring of food waste is a key prerequisite for any concerted action to reduce it. A food waste index is under development by United Nations Environment Programme. [30]. This agency positively assesses progress in global monitoring of food waste especially at a household level, with the most even distribution of results across income groups [6], which is confirmed also in the European scale [31]. OECD is another influential intergovernmental organization concerned about food waste [32]. Its statistical system reports household food waste amounts for its member states [7]. However, the compiled data come from various sources and times, and thus lack comparability. 

Several assessments have been made to obtain common data and information on people/households who waste food [33,34,35]. Despite this, experts and policymakers have only a rough notion of the food waste reality so far because some surveys use self-reported perceptions and opinions, while others use data and information that are hardly comparable due to methodological inconsistencies. Therefore, many countries have conducted their own data collection and analyses to set targets, design policies and support desired actions [36]. The Czech government claimed allegiance to key global sustainability issues in 2017 when it adopted the Sustainable Development Goals and thus also efficient, resource-saving and sustainable food management. As an EU-27-member state, it endorsed all the EU food-saving oriented policies and measures [37]. Reliable data on all parts of the food chain is a prerequisite for meeting these international reporting requirements and setting sound national policies. The main approaches for data collection are referenced in a further section. 

### 2.2. Food Wastage—The Current State 

Regarding food waste just at the household level, huge differences exist between countries. The average consumer in Europe and North America wastes 95–115 kg per year, while in subSaharan Africa and South/Southeast Asia this figure is 6–11 kg per year. However, not only the data range markedly, but also the information is very heterogeneous due to a lack of common methodologies for gathering food waste data. At the EU level, the latest available household food waste survey showed that EU-27 citizens waste 76 kg of edible food per capita a year. The lowest value of 25 kg per capita was observed in the Czech Republic and Slovakia, and the highest values of 113 and 133 kg per capita in Luxembourg and the UK, respectively [33,38]. The highest share of people (80%) wasting no or little food (up to 5%) also reside in the Czech Republic. This was revealed by two Flash Eurobarometer surveys (No. 316 in 2011 and No. 388 in 2013), which investigated citizens’ perceptions and practices related to resource efficiency and waste management [39]. The surveys provided individuals’ estimation at the EU-27 level concerning the share of food wasted through self-administrated questionnaires [40,41]. Another survey (No. 425 in 2015) focused on citizens’ attitudes towards the prevention of food waste, i.e., the responsibility of specific actors and actions helping citizens to reduce the amount of food waste in their own homes. Most Europeans accepted individual responsibility for food waste, with nearly two-thirds saying that better shopping and meal planning would contribute to the reduction of waste [39].

Other studies provide similar results. The recent pan-European FUSIONS study estimated 88 million tons of food waste per year, amounting to 143 billion euros. The sector contributing the most to EU food waste is households at 47 million tons, including both edible and inedible parts. This is equivalent to 92 kg per person per year; however, data from only eleven EU member states were extrapolated to give a total figure, and thus the results have relatively high uncertainties. The data on food waste in the Czech Republic was not included because it was not available or of low quality, and food waste in households was not analyzed because insufficient information was appropriate for processing [42]. Similarly, a study on food waste across the EU-27 did not include data on food waste by Czech households, although some sectors were covered [43]. 

So, seemingly, there are some data, but they are loaded with high uncertainties and incomparability. Bräutigam, Jörissen and Priefer compared different calculation methods and the reliability of their results across the EU-27 [44]. After analyzing both major panEuropean studies and several national studies, they obtained hardly any comparable data and making a solid estimation of the extent of food waste was difficult. Quantities were estimated, referred to different sources, used various methods for data collection, and most national studies extrapolated results from random waste analyses at the district and regional level to a whole country. Nevertheless, the total amount of 254 thousand tons in total, and 25 kg per capita a year of food waste by the Czech households in 2006 and 2009, served as a starting point and a reference for results obtained in the study. Similarly, based on the best available data of different origins (EUROSTAT, national studies, and the minimum scenario calculated as a share of organic waste from municipal residual waste), an estimated EU-27 benchmark of 76 kg food waste per capita a year may be considered [33,44].

### 2.3. Food Waste Monitoring

The European Union, committed to halving food waste by 2025 [45], recently issued decisions establishing a common methodology and determining the minimum quality requirements for food waste data collection [22,46]. It is clearly stated that national food waste reduction strategies and reporting should be based on reliable data updated every four years. While some food waste streams fall under voluntary measurement activities (e.g., member states can measure and provide the Commission with data on food waste drained as/with wastewaters voluntarily, which would give a more complete picture of the food waste situation), the main household food waste flows will be measured by any, or by a combination of, the following methods. 

Direct methods that directly quantify the amount of food waste by weighing, waste composition analysis (WCA), surveys, diaries, records, and observation.Indirect methods, where food waste is estimated from various secondary data sources such as modeling, mass balance, proxy data and literature data.

The Joint Research Centre of the EU conducted a review of the methods used by EU member states and found a wide range of measurement methods and, in most cases, more than one method was used within the same study [31]. The European Commission suggests methods of measurement for each stage of the food chain [46]. Table 1 shows direct methods in the green-shaded cells and indirect methods in the blue-shaded cells, whilst yellow-shaded cells include both direct and indirect methods. To increase the robustness of food waste results, the most accurate methods should be used in the quantification of food waste [47]. Additionally, more than one direct method can be used on the same sample to assess the robustness of the results, as performed by Giordano et al. who triangulated results obtained through WCA, surveys and diaries [48].

Every measurement method quantifying food wastage has pros and cons related to its objectives, and these might influence significantly the results obtained. In general, physical waste surveys are considered more objective and accurate than self-reporting methods, as they are carried out by a third party (usually a subcontractor of the local authority) with expertise in field waste surveys. [49]. As far as reporting methods are concerned, it is known that questionnaires underestimate the amount remarkably, while diaries, photo records and the use of receptacles (weighing or assessing volume at regular intervals) give similar, and slightly underestimated, numbers on food waste to each other. Nevertheless, food waste diaries are an important tool for monitoring the wasting of food, in particular for understanding the types, reasons and routes of discarded food. In terms of quantification, when an interpretation of results is done, it is necessary to bear in mind that diaries underestimate the amount of household food waste by 7–40% [50]. 

The most important finding of the recent review studies shows that there is a significant lack of food waste data based on direct measurements at all stages of the food supply chain. Both the European survey [31] and the global overview examining 202 relevant publications from 84 countries [51], identified literature and proxy data combined as the main sources of data. Only 20% of the studies were based on primary data, which signals high uncertainties in the existing global food losses and waste inventories. Therefore, the authors call for more fieldwork and primary data collection to fill data gaps and help to verify existing estimations, and thus contribute to a more accurate and reliable picture of food waste generation.

The European-funded food waste prevention project, FUSIONS, recently published updated food waste estimates for the EU-28 and a Food Waste Quantification Manual. The Manual is to support the EU member states in establishing more reliable monitoring and reporting of national food waste data at each stage in the food supply chain. This research showed that there are still some major data gaps preventing more comprehensive and reliable assessments of food waste levels in the EU [42]. 

## 3. Data and Methodology

There are several methods for collecting data about food waste (see Section 2). As we preferred a feasible (in terms of required resources), internationally recognized method provided reliable results, we decided on the kitchen diary methodology [52,53] The kitchen diaries method was designed to obtain information about the amount and type of food waste, disposal routes and reasons for disposal, as well as the sociodemographic characteristics of Czech households. 

The survey was conducted in 403 households (958 people) resulting in 403 fully completed questionnaires. Data collection used computer-assisted web interviewing (CAWI) via an online panel in October 2017. Participants were selected by quota nonprobabilistic sampling. The sample representativeness was ensured by defining quotas derived from the real distribution of the required characteristics in the population of the Czech Republic. The monitored quotas were the size of the residence area (corresponding to the settlement structure of the Czech Republic), household size and housing type (apartment in an apartment building, house). Proportionality of the overall distribution within the regions of the Czech Republic was also secured. Thus, the data from this survey can be considered representative for the population of the Czech Republic. The differences in the percentage of defined individual categories of the data set compared to the real distribution are around one percentage point on average. 

Supervision and coordination of the data collection were provided by the well-known company Median Ltd., which has extensive and long experience with sociological surveys. Participants (household members) were provided with full instructions on the use of kitchen diaries with emphasis on recording all food waste (edible and inedible parts), food waste thrown away by all people in the household, for any reason, no matter where the food was thrown. During the survey, attention was paid to reducing bias in household data collection as much as possible by the following actions.

Explaining to participants the anonymity of data collection, thus preventing the identification of the household.Providing e-mail and telephone support to participants during household data collection to help them record the data immediately (e.g., help them determine what measurement methods should be used or how to record the food waste properly).Supporting participants by online communication to remain in contact with them, maintain motivation, and to remind them to regularly complete diaries with data.

The households recorded every food waste item for one week (seven consecutive days). According to recommendations, participants were advised to use a measuring scale (for weight) or a measuring jug (for volume), but they were also allowed to record the quantity by counting (the number of items = 1 apple), tablespoons and 3-dl cups [54,55]. Not having to weigh all items might have caused some measurement inaccuracies (by conversion to weight) but greatly reduced the burden of diary-keeping and gave some flexibility to diary holders.

Besides the food waste flows, the diaries also recorded information difficult to obtain by any other method, such as the reasons for food disposal and the main disposal routes from home (e.g., flushing to the sewer from the kitchen sink or toilet, composting or feeding animals). Participants had to record each item of food thrown away as accurately as possible (e.g., not ‘banana’ but ‘banana peel’). 

Descriptions of the food or drink waste items from diaries were subsequently coded into food types (e.g., banana, carrot, bread) and grouped into food categories (fruit, vegetable, bakery) by the research team. The definition of food categories was based on the WRAP methodology [52] but adapted to national conditions and low values in some categories. Some categories were clustered (fresh + processed fruit, fresh + processed vegetables, cake and desserts + confectionery and snacks). Furthermore, the data on quantities (volume, the number of items, tablespoons, 3dl cups) were converted to weight. To calculate the weights of the food items’ edible and inedible proportions, data on the composition of food from a national source were used [56]. This database includes coefficients on the edible proportions of particular food items, e.g., an apple has a coefficient on the edible proportion of 0.92, which means that 92% of the apple weight is considered edible and 8% inedible.

In reviewed studies, the terms edible–inedible food waste and avoidable–unavoidable food waste have often been used interchangeably. Since “avoidable food waste” comprehensibly denotes the pointlessness of the generation of this particular food waste type, we use the division into avoidable and unavoidable food waste in the results section as follows. 

Avoidable food waste is food and drink thrown away that was, at some point prior to disposal, edible (This category also includes potentially avoidable food waste, e.g., bread crusts and potato skins).Unavoidable food waste is waste arising from food or drink preparation in households that is not, and has not been, edible under normal circumstances (e.g., meat bones, eggshells, pineapple skin, tea bags) [57].

After cleaning the data (errors and consistency checks), data were analyzed using descriptive statistics. The selected results were finally extrapolated to one year for the whole Czech population, and values per person and household were calculated. A flowchart with methodological steps is shown at Figure 1.

## 4. Results

During the one-week measurement period, the surveyed households produced 1051.5 kg of food waste. On average, surveyed households discarded 2.61 kg (1.10 kg per capita weekly). After extrapolation, the total food waste was estimated to be 135.68 kg per household or 57.08 kg per capita annually (Table 2).

Avoidable food waste accounted for 59% of the total amount of food waste produced in the surveyed households. The unavoidable part of food waste (41%) was mostly made up of the inedible parts, vegetables and fruit (peels, stalks, woody parts, seeds), meat bones and teabags. Within food categories, the largest contributions to avoidable food waste were from vegetables (26%), bakery (23%), home-made meals (16%), and fruits (13%). Meat accounted for 7%, and dairy and eggs 6%. These six food categories altogether accounted for 91% of the total avoidable food waste (Table 3). From the results, it is obvious that most discarded food was perishable and leftovers from prepared home-made meals.

By extrapolation of the data to the total population of Czechia, we can state that in total Czechs waste about 355 thousand tons of avoidable waste parts of food yearly. This amount of food waste would fill up about 100,000 small trucks. If these were lined up close together on a road, they would occupy the 650 km-long route from Prague to Hamburg in Germany. If we take a closer look at the particular food types within the food waste categories, the most discarded type was bread (65% of the bakery category, 14.6% of the total avoidable food waste). To illustrate, when converting this to annual values, Czechs throw away 68 million loaves of bread per year. The second most discarded item among home-cooked meals was soup (59% of this category, 9.4% of the total avoidable food waste). After the same conversion, this is over 96 million servings of soup per year. The large wastage of soups can be likely explained by the traditionally high level of soup consumption in Czechia [58].

Figure 2 shows that almost half the avoidable food waste was classified as “not used in time, spoiled”. The other half was spread across six reasons for wastage, dominated by “cooked, prepared or served too much” (17%).

The amount of discarded food in the surveyed households naturally varies commensurate with the different numbers of members. It increases with increasing members in the range 1–5 + (Table 4). The trend is growing, but for a household with three and four members, the total amount and the unavoidable amount of food waste are similar. However, the results in (Table 4) show that the higher the number of household members, the lower the amount of food waste per person (apart from avoidable food waste in two-member households, which slightly exceeds the amount of avoidable food waste in one-member households). Thus, one-member and two-member households waste the most (in terms of per capita results) and, at the same time, these households make up the majority (over 60%) of all households. 

This seems to be a common problem in Denmark. Single households make up 37% of all households and create the most food waste when compared to households with several members. Due to high environmental awareness, and possibly also economic reasons, about 60% of single Danes demand smaller packages in supermarkets [27]. Unlike in Denmark, food over-shopping concerns all household types (in terms of the size) in our research. This is caused by the fact that the Czechs are used to buying discounted food or food in large cost-effective packages. Low price, then, is a trigger for buying more than needed. These findings are observed by regular opinion polls related to food consumption [8].

We also looked at the differences in food waste according to the size of residence (Table 5). In all cities of various sizes, the average person throws out a similar amount of food waste, with a maximum difference of just 6%. Prague as an extraordinary category, being the capital and the largest city in the country, deviates from this trend. The average Prague inhabitant throws away 20–25% less than inhabitants of any smaller municipality. We can speculate and explain this deviation by the different lifestyle of Prague’s inhabitants; in general, people working in banks, state administration and tourism, tend to prepare less food at home (long working hours, high income, and a high supply of restaurants and fast food) [59]. 

Households also recorded disposal routes, i.e., where individual items of food waste are thrown. Half of the total food waste (540 kg, 51%) from surveyed households was thrown into municipal residual waste bins (for mixed waste from households that cannot be recycled). Another amount of 223 kg was fed to animals, and a similar amount of 210 kg was composted. The final 79 kg was disposed of via the sink, toilet or another outlet to the sewer system (Table 6). From the environmental impact point of view, these results are quite negative, in that most of the measured waste ends up where it should not. In concrete numbers, 59% of the total food waste (619 kg) ends up in municipal residual waste or sewage. After extrapolation, Czechs dispose of 357 thousand tons of food waste into the municipal residual waste or sewage annually. The waste statistics show that 46% of municipal residual waste in Czechia goes to landfills where food waste contributes to landfill gas with a high proportion of methane [60]. Municipal residual waste for energy use cannot contain food waste because it reduces the calorific value of incinerated waste [61]. Another undesirable route is into the sewer system due to food waste impacts on water consumption, the sewerage system and wastewater treatment processes [62]. The research also showed that 41% of the total food waste (432 kg) was composted or fed to animals, which may be considered the best way to dispose of food waste in the home environment if households do not have the option of a separately collected food waste container (which is quite exceptional in Czechia). This high share may be justified by the very fragmented settlement structure in Czechia, with a large share of small municipalities with houses having gardens, and often keeping domestic animals. Composting and feeding food waste to animals has been a long tradition in these areas. Unfortunately, there are no other Czech studies for comparison, and further research will be needed to confirm our results. On the other hand, it is also likely that the result of 41% of the total food waste composted or fed to animals is extremely overestimated. A British study reported a much lower value of only 11% of the total food waste being composted or fed to animals [52]. Again, any international comparison must be made with caution because of methodological discrepancies, e.g., the British results were a synthesis of several data collection methods (kitchen diaries, food waste compositional data, and detailed waste compositional analysis). 

If we look at the disposal route within the size of the residence, food waste (per average person) composted and fed to animals decreases with the increasing size of municipalities. In municipalities with 10,000 or more inhabitants, the amount is the same. Conversely, the amount of food waste in municipal residual waste bins increases with the increasing size of municipalities (with irregularity in the two categories with the highest population) (Table 7). If we express the amount of food waste per person, in particular disposal routes as a share of the total food waste per person in categories of the municipality size, it turns out that the average person in Prague throws away 34% more waste into the municipal residual waste bin than in municipalities with less than 1999 inhabitants. This amount (34%) in municipalities up to 1999 inhabitants is thrown into compost, or given to animals, instead deposited in a municipal residual waste bin. We can deduce that if composting opportunities improved in larger municipalities, people would not throw so much food waste into the municipal residual waste. In municipalities with over 10,000 inhabitants, more than 60% of food waste is discarded into municipal residual waste bins.

Interesting results were also provided by housing type split by disposal route. The average person living in a house puts about 60% more food waste into compost, or feeds to animals, than the average person living in an apartment (in an apartment building). In contrast, an average person living in an apartment throws almost half as much into municipal residual waste and 43% more into the sewer system than the average person from a house. 

Eighteen households, i.e., 4% of a total of 403 households, reported no food waste at all. These households had the following characteristics: they were mainly small households (10 one-member households and five two-member households), most had a low monthly income and almost half of these households had senior citizens. When recording that they did not throw away any food that day, they usually stated the reason “all the food was eaten/there was no reason to throw away the food” or “they did not eat anything at home this day”. Due to the characteristics of the no-wastage households, we believe the data to be plausible, and they were not excluded from the analysis.

## 5. Discussion and Conclusions

This study’s results on food wastage by Czech households will contribute to the national experience on this type of research and fill the gap in EU and international statistics. With caution, regarding international comparison, the total food waste per person per year of 57 kg is slightly lower than in the other new EU member states with similar socioeconomic and political characteristics (Hungary has 68 kg of the total food waste per person a year) also applying the kitchen diary approach. The avoidable part of Hungarian food waste is quite similar to the Czech (Czechia 33 kg, Hungary 35 kg) [14]. 

A connection between food consumption and wastage in households certainly exists [63]. The Czech Statistical Office has had a continuous time series of data for the Czech Republic since 1948. Therefore, it is possible to characterize changes in eating habits, which can help to interpret some of the results of this study. The political, economic and socio-cultural changes of the 1990s (after the fall of communism in 1989) have had a significant effect on food consumption. Consumption has been affected by a number of factors, e.g., price liberalization associated with rising prices, the development of population incomes and their differentiation, the prices of substitute foods, the possibility of imports, the availability of food and products, and the reduction of self-sufficiency, have been crucial. Advertising and nutritional consultants have played an important role. Food consumption has also been affected by changes in eating habits, lifestyle and tourism. The total consumption of food in the Czech Republic (including food losses and waste) reached its maximum in 2019. Since the split of Czechoslovakia and the foundation of the independent Czech Republic in 1993, the total consumption of food per capita a year has increased by 65 kg to 796 kg (a year-on-year increase of almost 7 kg) [11]. Food consumption information provides an important context for further interpretation of this study’s results since it is a factor affecting the total food waste stream. 

As in some other countries, for example Finland [64,65], the largest food waste category was vegetables. After 1989, the consumption of vegetables and, depending on imports, also fruit, especially southern fruit, have been steadily rising in the Czech Republic [11]. Vegetables (and fruits as the fourth most wasted category) have become much more affordable financially, more freely available and have become a daily food for many people. At the same time, vegetables and fruits together have become by far the largest category of food waste (40% of the total wasted food). Increased affordability might lead to purchase in larger quantities, some of which will not be processed. This is indicated by the answer that the main reason for throwing away vegetables (and fruits) was primarily spoilage. The reason for this may be (unlike, for example, meat or dairy products) often ambiguous information about storage, such as in the refrigerator or outside, packaged, or loose. The second-largest category of food waste was bakery products. The large wastage in this category can be explained by the high level of consumption of bread and wheat bakery products in Czechia, which has been steadily increasing since 1950 [11]. Many people prefer fresh bakery products, and their relatively low price encourages waste. The third-largest category of food waste was food prepared at home comprised of raw ingredients. Home-cooked meals, however, require other precious resources than just raw ingredients, such as energy (for shopping, storing the raw ingredients, and cooking) and time, undoubtedly causing some negative environmental impacts. Meals were often thrown away primarily because they were cooked, prepared, or served in too large amounts and left. These leftovers were either thrown away immediately or spoiled later and then thrown away.

Adequacy in purchasing and cooking quantities is an important prerequisite for sustainable food management in households. In general, when people prefer buying large packages of food and food ingredients, a large part of the food ends up as waste. Discounts and favorable prices are often offered for large packages, such as XXL or economy/family size. Such inadequate shopping may, in particular, be a problem in one or two-person households where it can often become a source of food waste. However, surveys have revealed that over-shopping exists in a large portion of the Czech population. Thus, this problem is likely to be growing not only in Prague, which has the highest share of single households (nearly 40%) [66]. Besides adequate shopping, portion sizes also should be based on moderation and modern nutritional knowledge (while avoiding food waste) [67]. 

As regards disposal routes, Czech households mostly throw food away into municipal residual waste. The untapped potential for nutrient cycling and reducing the environmental impact of food waste is obvious. From this point of view, 41% of the total food waste composted or fed to pets and other domestic animals is a positive result (if validated by further research). Municipalities in Czechia are currently obliged to provide places for disposal of biodegradable waste (plant, vegetal) from gardens, but they mostly do not handle the issue of food waste. A possible solution could be to create infrastructure for food waste collection (e.g., a door-to-door system) and transport to a biogas plant. This already functions in some European cities as Milan, Malmo and Bristol [68,69,70]. Prague has been testing this method for two years [71], and some other Czech cities are ready to follow a good practice example. The development of a sustainable solution for food waste management in the Czech Republic is urgently needed because EU and national targets are certainly not being met, and the share of food waste still accounts for 20–40% of municipal residual waste ending up in landfills [72].

The chosen survey method of kitchen diaries enabled us to quantify the food waste stream as well as explore its important qualitative characteristics as disposal routes and reasons for disposal. Although the survey was scheduled only for one week, maintaining regular contacts with respondents by phone and/or email (resolving any relevant issue but mostly reminding them to fill in the diaries) proved especially important. As mentioned in the review section, diary-based research tends to report lower quantities of food waste. In the Data and Methodology section we pointed out that, among other reasons, the conversion of some food waste items to weight units might have caused some measurement inaccuracies. On the other hand, the possibility to choose from several options of units for recording food waste items reduced the diary-keeping burden remarkably and gave convenient flexibility to diary holders. We are aware that recording itself may also reduce food waste flows. As described in previous studies (see e.g. [49]), the food diaries may be influenced by social desirability bias whereby people change their waste-discarding habits or underreport their waste in order to present themselves in a positive light. However, this study was not able to quantify and assess uncertainties caused by the above factors. For any results comparisons (within the country as well as among countries), the surveyed method needs to be taken into account because only data loaded with the same/similar method-biases are comparable. Data triangulation can ensure validity of the collected data (e.g., data from garbology survey). However, such data do not exist yet. Therefore, development of another type of the household food waste survey could be the next step helping to confirm the real situation.

The overall research design fully complies with recommended standards, and the country thus contributes to the international effort on sustainable food management. Food as a complex category represents the most inherent component of human life, and should be approached by sociological, economic, ethnographic and environmental methods, and in various contexts. It is important to keep in mind that countries in postcommunist transformation may have, due to a long undemocratic period, specific features related to this research topic. Food and food wastage are an appropriate theme (or one of the themes) for exploration of postsocialism in central and eastern Europe to better understand prevailing consumption patterns, the role of education and awareness raising, as well as cultural and economic factors. Besides these potential directions of next steps, the focus of further research should extend from one sector (households) to other major parts along the food chain such as food services, both public and private (restaurants and pubs, cafes and canteens). 

## Figures and Tables

**Figure 1 foods-10-00875-f001:**
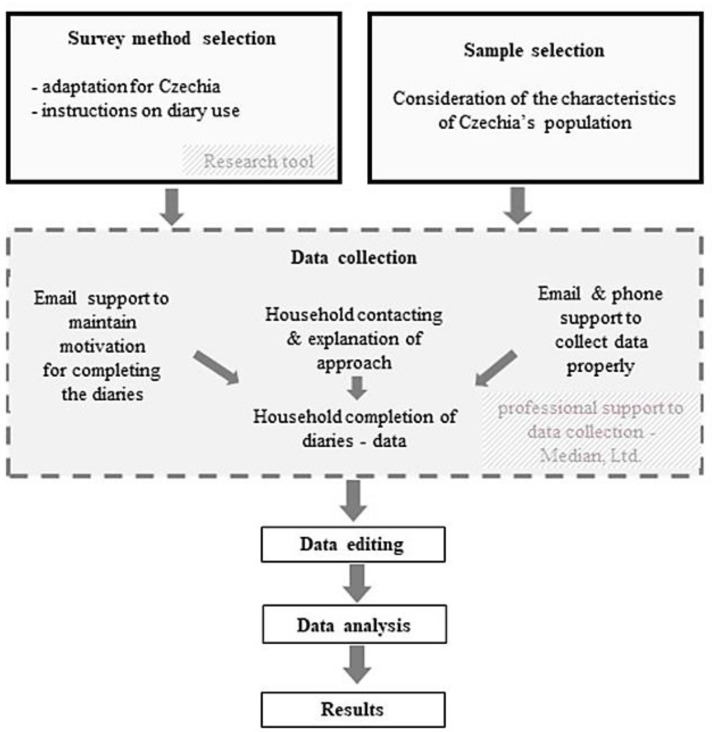
A flowchart with main methodological steps.

**Figure 2 foods-10-00875-f002:**
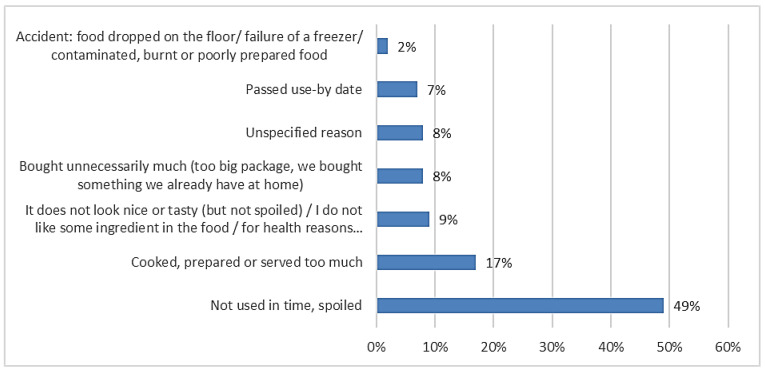
The proportion of avoidable food waste by reasons for disposal, based on the one-week measurement.

**Table 1 foods-10-00875-t001:** Accuracy and reliability (*** high, ** medium) of the food waste measurement methods suggested by the European Commission (adapted based on [46]).

Stage of the FCS	Methods for Food Waste Measurement					
Primary production	Direct measurement (weighing or volumetric) ***	Questionnaires and interviews **	Mass balance **		Coefficients and production statistics **	
Processing and manufacturing						
Retail and food distribution				Waste composition analysis ***	Counting/scanning **	Diaries **
Restaurants and food services						
Households						

**Table 2 foods-10-00875-t002:** Food waste during one-week measurement and estimation of annual values.

Food Waste in Surveyed Households per Week (403 Households, 958 Household Members)			Annual Calculated Food Waste of the Czech Population	Annual Calculated Food Waste per Household	Annual Calculated Food Waste of the Czech Population per Person
Type of food waste	Weight (kg)	Proportion	Weight (kg)		
Avoidable	615.72	59%	354,602,237	79.45	33.42
Unavoidable	435.77	41%	250,966,976	56.23	23.65
Total food waste	1051.5	100%	605,569,213	135.68	57.08

**Table 3 foods-10-00875-t003:** Quantity of avoidable food waste by food categories based on a one-week measurement.

Food Categories	Avoidable Food Waste in Surveyed Households per Week (kg)	Proportion
Vegetable	162.93	26.40%
Bakery	138.65	22.52%
Home-cooked meals	97.79	15.88%
Fruit	80.34	13.05%
Meat	42.29	6.87%
Dairy and eggs	37.42	6.08%
Staple foods (pasta, rice, buckwheat, flour, oatmeal, corn)	24.46	3.97%
Drinks	22.35	3.63%
Sweet and salty snacks	4.69	0.69%
Condiments, sauces, herbs, and spices	2.82	0.46%
Oil and fat	1.98	0.32%
Nuts and seeds	0.47	0.08%
Total	615.72	100.00%

**Table 4 foods-10-00875-t004:** Quantity of food waste by household size based on the one-week measurement.

Number of Household Members	Food Waste (kg)					
	Values per Household per Week			Values per Person in Household per Week		
	Total Amount	Avoidable Amount	Unavoidable Amount	Total Amount	Avoidable Amount	Unavoidable Amount
1	1.40	0.76	0.64	1.40	0.76	0.64
2	2.69	1.58	1.11	1.35	0.79	0.55
3	3.41	1.98	1.42	1.14	0.66	0.47
4	3.33	2.01	1.33	0.83	0.50	0.33
5+	5.80	3.60	2.20	0.78	0.48	0.30

**Table 5 foods-10-00875-t005:** Quantity of food waste by the residence size.

Size of Residence	Food Waste in Households per Person per Week (kg)		
	Total Amount	Avoidable Amount	Unavoidable Amount
up to 1999 inhabitants	1.10	0.66	0.43
2000–9999 inh.	1.17	0.75	0.42
10,000–49,999 inh.	1.10	0.59	0.52
50,000 or more inh. (excluding Prague)	1.15	0.61	0.54
Prague (1.3 mil. inh.)	0.88	0.55	0.33

**Table 6 foods-10-00875-t006:** Quantity of food waste by all household members per week by a disposal route.

Disposal Route	Food Waste in Surveyed Households per Week (kg)			Annual Calculated Food Waste of the Czech Population (Tonnes)
	Avoidable Amount	Unavoidable Amount	Total Amount	
Municipal residual waste	274.8	265.2	540	31,097
Sewer	74.1	5.4	79.4	45,744
Composting	88.3	121.2	210	120,678
Feed to animals	178.6	44	223	128,177

**Table 7 foods-10-00875-t007:** Quantity of households’ food waste by disposal route and residence size.

Size of Residence	Food Waste in Households per Person per Week (kg)			
	Composting	Municipal Residual Waste	Sewer	Feed to Animals
up to 1999 inhabitants	0.29	0.39	0.06	0.35
2000–9999 inh.	0.24	0.42	0.14	0.37
10,000–49,999 inh.	0.17	0.68	0.10	0.15
50,000 inh. or more (excluding Prague)	0.17	0.83	0.06	0.10
Prague (1.3 mil. inh.)	0.17	0.61	0.05	0.05

## Data Availability

The data presented in this study are available on request from the lead author.

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
