# Peer review of "An Analysis of Food Waste in Czech Households—A Contribution to the International Reporting Effort"

_foods, 2021, doi:10.3390/foods10040875_

Round 1
Reviewer 1 Report
Very interesting study conducted by Nováková et al. which analyzed the food waste in Czech households. I found a lot of important data and relevant results in this research. However I have some comments that I would like the authors take in consideration.
The methodological approach needs to be mentioned in the abstract.
“Czech Republic” should be added as a keyword.
The authors should mention and discuss the most recent dietary guidelines that take in consideration sustainability concerns:
Serra-Majem, L.; Tomaino, L.; Dernini, S.; Berry, E.M.; Lairon, D.; Ngo de la Cruz, J.; Bach-Faig, A.; Donini, L.M.; Medina, F.-X.; Belahsen, R.; Piscopo, S.; Capone, R.; Aranceta-Bartrina, J.; La Vecchia, C.; Trichopoulou, A. Updating the Mediterranean Diet Pyramid towards Sustainability: Focus on Environmental Concerns. Int. J. Environ. Res. Public Health 2020, 17, 8758. https://doi.org/10.3390/ijerph17238758
Fernandez, M.L.; Raheem, D.; Ramos, F.; Carrascosa, C.; Saraiva, A.; Raposo, A. Highlights of Current Dietary Guidelines in Five Continents. Int. J. Environ. Res. Public Health 2021, 18, 2814. https://doi.org/10.3390/ijerph18062814
I would opt to merge section 1 with section 2 and the text of the current sections 1 and 2 should be reduced. Then, the main goals of the present study should be clearly stated at the end of this Introduction section. All of this information needs to be summarized.
Section 3: “After considering the pros and cons of the data collection methods appropriate for the household level” – What are these pros and cons? Clarify this in the manuscript.
It would help the readers if you could provide a flowchart in section 3 with all the steps you made in the present study.
Study limitations should be pointed out by the authors at the end of Discussion. In my point of view, the Discussion section should be separated from Conclusion.
Further investigations should be stated by the authors.
Author Response
Dear Editor and Reviewers,
Thank you for very relevant comments and suggestions. We gladly accepted most of them and amended the text accordingly. The new additions and changes were checked by our English native proof-reader. We hope the manuscript meets the FOODS´s quality criteria now.
With best wishes,
Tomas Hak (on behalf of the authors team)
Author's Reply to the Review Report (Reviewer 1)
Very interesting study conducted by Nováková et al. which analyzed the food waste in Czech households. I found a lot of important data and relevant results in this research. However, I have some comments that I would like the authors take in consideration.
The methodological approach needs to be mentioned in the abstract.
Accepted. The method is clearly identified there that the readers interested in particular in the kitchen diaries methodology get this information at the very beginning.
“Czech Republic” should be added as a keyword.
Accepted. The keyword added.
The authors should mention and discuss the most recent dietary guidelines that take in consideration sustainability concerns:
Serra-Majem, L.; Tomaino, L.; Dernini, S.; Berry, E.M.; Lairon, D.; Ngo de la Cruz, J.; Bach-Faig, A.; Donini, L.M.; Medina, F.-X.; Belahsen, R.; Piscopo, S.; Capone, R.; Aranceta-Bartrina, J.; La Vecchia, C.; Trichopoulou, A. Updating the Mediterranean Diet Pyramid towards Sustainability: Focus on Environmental Concerns. Int. J. Environ. Res. Public Health 2020, 17, 8758. https://doi.org/10.3390/ijerph17238758
Fernandez, M.L.; Raheem, D.; Ramos, F.; Carrascosa, C.; Saraiva, A.; Raposo, A. Highlights of Current Dietary Guidelines in Five Continents. Int. J. Environ. Res. Public Health 2021, 18, 2814. https://doi.org/10.3390/ijerph18062814
Not accepted. The article deals with food wastage, not with food in terms of nutrition. Also, we are supposed to reduce the review part of the manuscript.
I would opt to merge section 1 with section 2 and the text of the current sections 1 and 2 should be reduced. Then, the main goals of the present study should be clearly stated at the end of this Introduction section. All of this information needs to be summarized.
Partially accepted. Both sections were significantly amended and shortened. We did not merge both sections since they cover distinct areas – Section 1 introduces the topic and provides a rationale for the study, and stresses the extraordinary situation in the country of post-communistic transformation while Section 2 provides relevant literature review (we shortened this section remarkably and kept only references relevant to the research question of this article). The research objectives are stated at the end of the Introduction section. The manuscript was reduced to a total of about 7,5 thousand words (excluding the list of references), i.e. a very standard format for the research article.
Section 3: “After considering the pros and cons of the data collection methods appropriate for the household level” – What are these pros and cons? Clarify this in the manuscript.
Accepted. We decided for a feasible (in terms of resources), internationally recognized method provided reliable results. Therefore, we decided for a direct method – kitchen diaries – which slightly underestimates the results but on the other hand, it allows for understanding the types, reasons, and routes of discarded food. Text amended in this sense.
It would help the readers if you could provide a flowchart in section 3 with all the steps you made in the present study.
Accepted. A methodology-illustrating flowchart inserted in Section 3.
Study limitations should be pointed out by the authors at the end of Discussion. In my point of view, the Discussion section should be separated from Conclusion.
Partially accepted – we replaced the limitations passage to the end of the final Section. However, we did not separate the Discussion and Conclusion sections since they are closely interrelated. In many cases research papers use this design, other use even different structures (just the latest papers in the FOODS journal merge “Results and Discussion”, or have only Discussion section without any Conclusions, etc.).
Further investigations should be stated by the authors.
Accepted. A passage on the next steps added at the end of the final Section.
Reviewer 2 Report
Thank you for your manuscript. I suggest below a number of changes to improve the paper.
Abstract:
Please remove “From the point of view of environmental impact”, as there is not a reference to environmental impact in this sentence.
Section 1:
Second paragraph: please define low hunger “as defined by the Global Hunger Index”.
“one-third of the total food the world produces”: please cite http://www.fao.org/3/mb060e/mb060e00.pdf. You should also cite https://www.unep.org/resources/report/unep-food-waste-index-report-2021, which provides a more updated estimate.
Third paragraph: you may refer to only Czech Republic’s food waste generation when talking about “there is little data and information about the issue”, but because of the sentence “as likely most other industrialized countries”, it looks like the claims in this paragraph refer to all industrialized countries. I do not agree this paragraph applies to other industrialized countries, like the UK (see WRAP’s work).
I do not see the value of the three assumptions described in the numbered bullet points. As you disclose, they are “unsubstantiated assumptions”, which are furthermore contradictory. If you are basing the design of your study on a specific hypothesis to be tested, please explain only this one.
Section 2:
“in developed countries loss-waste mostly occurs in the final stages”: this is commonly referred to as “waste” only, not “loss” (which refers to the first stages of the supply chain only). I see this is mentioned two paragraphs below: these two paragraphs could be combined as they both refer to food loss and waste. Please also define these two terms.
First paragraph in Section 2.1: I refer again to https://www.unep.org/resources/report/unep-food-waste-index-report-2021, which give updates estimations and provides an alternative conclusion on this.
“Table 1 shows direct methods in the green-shaded cells and indirect methods in the blue-shaded cells, whilst yellow-shaded cells include both direct and indirect methods.”: however the table doesn’t show these colours. I am also not convinced about the accuracy of the table, e.g. questionnaires and interviews can also be used for household analysis, composition scanning could also be used in processing and manufacturing... Therefore, I suggest removing the table and explaining the most important aspects in text.
“it is known that questionnaires underestimate the amount remarkably”: this should be demonstrated or referenced.
Last paragraph: I would not say that FUSIONS is the latest initiative in this regard. FUSIONS finished in 2016 (and it started in 2012), while there are other initiatives still ongoing.
In general, I find Sections 1 and 2 too long. They just described very well-known facts and initiatives which will not be new to the reader. These sections should just provide an introduction to the topic, context and justification of the research. Because of this, I suggest removing most of Section 1 (particularly the Maslow’s Hierarchy of Needs, which is not related to this topic; global numbers around food waste, which are discussed later in Section 2; changes in the Czech Republic’s economy since the fall of communism, which is not related to this topic; and the three “assumptions”, as discussed above).
I also miss some relevant recent research. Apart from what I mentioned above, please refer to the work done in the Joint Research Centre of the European Commission in Ispra (Italy), for example: https://ec.europa.eu/jrc/en/publication/review-studies-food-waste-accounting-member-state-level, https://www.sciencedirect.com/science/article/pii/S0921344921000331, https://www.sciencedirect.com/science/article/pii/S0921344919302721, https://www.sciencedirect.com/science/article/pii/S2211912417301530. Important work regarding food waste reporting standards: https://flwprotocol.org/flw-standard/. This work gives very recent estimates of food waste in the UK, for comparison: https://www.sciencedirect.com/science/article/pii/S2352550920314202. Another relevant work: https://www.sciencedirect.com/science/article/pii/S0301479718313021. This work could also be relevant in terms of types of food waste and the treatments each can go through: https://link.springer.com/content/pdf/10.1007/s12649-016-9720-0.pdf.
Section 3:
“After considering the pros and cons of the data collection methods appropriate for the household level we decided on the kitchen diary methodology”: what were the pros and cons to make this decision?
Please remove unscientific phrases such as “It is hard to imagine this amount and one might think that such a small country cannot be so lavish”.
Section 5:
Please explain the limitations of the research, particularly how the results may be affected by the study participants, e.g. reporting lower food waste levels than in reality (to “make them look good”), forgetting to add in the kitchen diary a food that was wasted, the fact that participating in such study may influence them to reduce food waste so results cannot be extrapolated to all the country, perhaps study participants agreed to participate in the study because they were already more committed to reducing food waste, etc.
Author Response
Dear Editor and Reviewers,
Thank you for very relevant comments and suggestions. We gladly accepted most of them and amended the text accordingly. The new additions and changes were checked by our English native proof-reader. We hope the manuscript meets the FOODS´s quality criteria now.
With best wishes,
Tomas Hak (on behalf of the authors team)
Thank you for your manuscript. I suggest below a number of changes to improve the paper.
Abstract:
Please remove “From the point of view of environmental impact”, as there is not a reference to environmental impact in this sentence.
Accepted. The text removed.
Section 1:
Second paragraph: please define low hunger “as defined by the Global Hunger Index”.
Accepted. Definition of GHI inserted.
“one-third of the total food the world produces”: please cite http://www.fao.org/3/mb060e/mb060e00.pdf. You should also cite https://www.unep.org/resources/report/unep-food-waste-index-report-2021, which provides a more updated estimate.
Accepted. A new recent citation inserted.
Third paragraph: you may refer to only Czech Republic’s food waste generation when talking about “there is little data and information about the issue”, but because of the sentence “as likely most other industrialized countries”, it looks like the claims in this paragraph refer to all industrialized countries. I do not agree this paragraph applies to other industrialized countries, like the UK (see WRAP’s work).
Accepted. The changed text refers to „many other industrialized countries“ contributing to environmental impacts caused by food wasting. A lack of data and information then refers only to Czechia.
I do not see the value of the three assumptions described in the numbered bullet points. As you disclose, they are “unsubstantiated assumptions”, which are furthermore contradictory. If you are basing the design of your study on a specific hypothesis to be tested, please explain only this one.
Partially accepted. Despite we explicitly state that the research objective was not testing any hypotheses (but collecting first data on household food waste), we removed „assumptions“ since that term may imply testing hypotheses. We consider this paragraph important since it provides contextual information pointing out to the fact that countries in post-communistic transformation have different starting points, little knowledge on all underlying factors affecting consumer behaviours etc. We stress this point also at the end where talking about future steps.
Section 2:
“in developed countries loss-waste mostly occurs in the final stages”: this is commonly referred to as “waste” only, not “loss” (which refers to the first stages of the supply chain only). I see this is mentioned two paragraphs below: these two paragraphs could be combined as they both refer to food loss and waste. Please also define these two terms.
Accepted. The text changed appropriately (it was a mistake). The term „food loss“ is used occasionally – and when appropriate – in the literature review part (e.g. SDG Target 12.3).
First paragraph in Section 2.1: I refer again to https://www.unep.org/resources/report/unep-food-waste-index-report-2021, which give updates estimations and provides an alternative conclusion on this.
Accepted. We added info on monitoring progress particularly positive in the household sector (page 4; citation UNEP, 2020).
“Table 1 shows direct methods in the green-shaded cells and indirect methods in the blue-shaded cells, whilst yellow-shaded cells include both direct and indirect methods.”: however the table doesn’t show these colours. I am also not convinced about the accuracy of the table, e.g. questionnaires and interviews can also be used for household analysis, composition scanning could also be used in processing and manufacturing... Therefore, I suggest removing the table and explaining the most important aspects in text.
Accepted partially. The table format is a problem of the MDPI Editorial system – originally, the table had a very different format (we will ask the editor to take care of). We agree with incompleteness/incorrectness of this resource, therefore we improved it and cite it as „adapted“ now.
“it is known that questionnaires underestimate the amount remarkably”: this should be demonstrated or referenced.
Accepted but nothing changed. This statement is (and was) appropriately referenced (Quested, T.E., Palmer, G., Moreno, L.C., McDermott, C., Schumacher, K. Comparing diaries and waste compositional analysis for measuring food waste in the home. Journal of Cleaner Production 2020, 262, p.121263)
Last paragraph: I would not say that FUSIONS is the latest initiative in this regard. FUSIONS finished in 2016 (and it started in 2012), while there are other initiatives still ongoing.
Accepted. The text changed.
In general, I find Sections 1 and 2 too long. They just described very well-known facts and initiatives which will not be new to the reader. These sections should just provide an introduction to the topic, context and justification of the research. Because of this, I suggest removing most of Section 1 (particularly the Maslow’s Hierarchy of Needs, which is not related to this topic; global numbers around food waste, which are discussed later in Section 2; changes in the Czech Republic’s economy since the fall of communism, which is not related to this topic; and the three “assumptions”, as discussed above).
Mostly accepted. Both sections were significantly amended and shortened. The manuscript was reduced to a total of about 7,5 thousand words (excluding the list of references), i.e. a very standard format for a research article. The contextual information on the Czech post-communistic transformation is relevant for the paper (despite the country belongs to the high-income countries, it has many specifics related to the research subject).
I also miss some relevant recent research. Apart from what I mentioned above, please refer to the work done in the Joint Research Centre of the European Commission in Ispra (Italy), for example: https://ec.europa.eu/jrc/en/publication/review-studies-food-waste-accounting-member-state-level, https://www.sciencedirect.com/science/article/pii/S0921344921000331, https://www.sciencedirect.com/science/article/pii/S0921344919302721, https://www.sciencedirect.com/science/article/pii/S2211912417301530. Important work regarding food waste reporting standards: https://flwprotocol.org/flw-standard/. This work gives very recent estimates of food waste in the UK, for comparison: https://www.sciencedirect.com/science/article/pii/S2352550920314202. Another relevant work: https://www.sciencedirect.com/science/article/pii/S0301479718313021. This work could also be relevant in terms of types of food waste and the treatments each can go through: https://link.springer.com/content/pdf/10.1007/s12649-016-9720-0.pdf.
Accepted. Thanks a lot for these relevant resources. Due to the text reduction requirement, we added the most relevant to the manuscript.
Section 3:
“After considering the pros and cons of the data collection methods appropriate for the household level we decided on the kitchen diary methodology”: what were the pros and cons to make this decision?
Accepted. The text phrase was removed and the reasons for the particular method selection stated.
Please remove unscientific phrases such as “It is hard to imagine this amount and one might think that such a small country cannot be so lavish”.
Accepted. The phrase removed.
Section 5:
Please explain the limitations of the research, particularly how the results may be affected by the study participants, e.g. reporting lower food waste levels than in reality (to “make them look good”), forgetting to add in the kitchen diary a food that was wasted, the fact that participating in such study may influence them to reduce food waste so results cannot be extrapolated to all the country, perhaps study participants agreed to participate in the study because they were already more committed to reducing food waste, etc.
Accepted. The passage on research limitations added in the final Section (at the end before the text on next steps).
Round 2
Reviewer 1 Report
I still believe that it would be pertinent to include the discussion of the following papers:
Serra-Majem, L .; Tomaino, L .; Dernini, S .; Berry, E.M .; Lairon, D .; Ngo de la Cruz, J .; Bach-Faig, A .; Donini, L.M .; Medina, F.-X .; Belahsen, R .; Piscopo, S .; Capone, R .; Aranceta-Bartrina, J .; La Vecchia, C .; Trichopoulou, A. Updating the Mediterranean Diet Pyramid towards Sustainability: Focus on Environmental Concerns. Int. J. Environ. Res. Public Health 2020, 17, 8758. https://doi.org/10.3390/ijerph17238758
Fernandez, M.L .; Raheem, D .; Ramos, F .; Carrascosa, C .; Saraiva, A .; Raposo, A. Highlights of Current Dietary Guidelines in Five Continents. Int. J. Environ. Res. Public Health 2021, 18, 2814. https://doi.org/10.3390/ijerph18062814
The new third dimension of the Mediterranean Pyramid emphatically addresses the issue of food waste.
Author Response
I still believe that it would be pertinent to include the discussion of the following papers:
Serra-Majem, L .; Tomaino, L .; Dernini, S .; Berry, E.M .; Lairon, D .; Ngo de la Cruz, J .; Bach-Faig, A .; Donini, L.M .; Medina, F.-X .; Belahsen, R .; Piscopo, S .; Capone, R .; Aranceta-Bartrina, J .; La Vecchia, C .; Trichopoulou, A. Updating the Mediterranean Diet Pyramid towards Sustainability: Focus on Environmental Concerns. Int. J. Environ. Res. Public Health 2020, 17, 8758. https://doi.org/10.3390/ijerph17238758
Fernandez, M.L .; Raheem, D .; Ramos, F .; Carrascosa, C .; Saraiva, A .; Raposo, A. Highlights of Current Dietary Guidelines in Five Continents. Int. J. Environ. Res. Public Health 2021, 18, 2814. https://doi.org/10.3390/ijerph18062814
The new third dimension of the Mediterranean Pyramid emphatically addresses the issue of food waste.
Accepted - we inserted the first suggested reference (Mediterranean Pyramid) that is quite relevant
Also, we checked the MS for misspellings or mistakes.
Reviewer 2 Report
Thank you for the revised version of your manuscript. I believe it has been substantially improved. At this point, I just have a couple of minor comments to make:
- I still don’t find much value about the three “expected results” between pages 2 and 3, but I understand the authors consider them important. In any case, I would not consider them to have “the same likelihood”, even if the three of them are actually possible.
- Section 2.1: change “Food loss in the consumption stage” for “Food waste in the consumption stage”. As discussed, food loss only occurs in earlier stages of the supply chain (and this is also sometimes referred to as food waste, anyway).
- Although some study limitations are now mentioned in the last section, I would recommend slightly expanding this and include additional considerations such as those mentioned in the last comment of the last review round.
Author Response
I still don’t find much value about the three “expected results” between pages 2 and 3, but I understand the authors consider them important. In any case, I would not consider them to have “the same likelihood”, even if the three of them are actually possible.
Accepted - the text changed (the problematic wording removed)
Section 2.1: change “Food loss in the consumption stage” for “Food waste in the consumption stage”. As discussed, food loss only occurs in earlier stages of the supply chain (and this is also sometimes referred to as food waste, anyway).
Accepted - the text changed according to the suggestion
Although some study limitations are now mentioned in the last section, I would recommend slightly expanding this and include additional considerations such as those mentioned in the last comment of the last review round.
Accepted - the study limits were expanded by the social desirability bias and the relevant reference was inserted